# The Investigation of Associations between *TP53* rs1042522, *BBC3* rs2032809, *CCND1* rs9344, *EGFR* rs2227983 Polymorphisms and Breast Cancer Phenotype and Prognosis

**DOI:** 10.3390/diagnostics11081419

**Published:** 2021-08-05

**Authors:** Justina Bekampytė, Agnė Bartnykaitė, Aistė Savukaitytė, Rasa Ugenskienė, Erika Korobeinikova, Jurgita Gudaitienė, Elona Juozaitytė

**Affiliations:** 1Oncology Research Laboratory, Oncology Institute, Lithuanian University of Health Sciences, LT-50161 Kaunas, Lithuania; agne.bartnykaite@lsmuni.lt (A.B.); aiste.savukaityte@lsmuni.lt (A.S.); rasa.ugenskiene@lsmuni.lt (R.U.); 2Department of Genetics and Molecular Medicine, Hospital of Lithuanian University of Health Sciences Kaunas Clinics, LT-50161 Kaunas, Lithuania; 3Department of Oncology and Hematology, Hospital of Lithuanian University of Health Sciences Kaunas Clinics, LT-50161 Kaunas, Lithuania; erika.korobeinikova@lsmuni.lt (E.K.); jurgita.gudaitiene@lsmuni.lt (J.G.); elona.juozaityte@lsmuni.lt (E.J.)

**Keywords:** breast cancer, SNP, *TP53*, *BBC3*, *CCND*, *EGFR*, associations, phenotype, prognosis

## Abstract

Breast cancer is one of the most common oncological diseases among women worldwide. Cell cycle and apoptosis—related genes *TP53*, *BBC3*, *CCND1* and *EGFR* play an important role in the pathogenesis of breast cancer. However, the roles of single nucleotide polymorphisms (SNPs) in these genes have not been fully defined. Therefore, this study aimed to analyze the association between *TP53* rs1042522, *BBC3* rs2032809, *CCND1* rs9344 and *EGFR* rs2227983 polymorphisms and breast cancer phenotype and prognosis. For the purpose of the analysis, 171 Lithuanian women were enrolled. Genomic DNA was extracted from peripheral blood; PCR-RFLP was used for SNPs analysis. The results showed that *BBC3* rs2032809 was associated with age at the time of diagnosis, disease progression, metastasis and death. *CCND1* rs9344 was associated with tumor size, however an association resulted in loss of significance after Bonferroni correction. In survival analysis, significant associations were observed between *BBC3* rs2032809 and OS, PFS and MFS. *EGFR* rs2227983 also showed some associations with OS and PFS (univariate Cox regression analysis). However, the results were in loss of significance (multivariate Cox regression analysis). In conclusion, *BBC3* rs2032809 polymorphism was associated with breast cancer phenotype and prognosis. Therefore, it could be applied as potential markers for breast cancer prognosis.

## 1. Introduction

Breast cancer is one of the most common cancers and the second leading cause of cancer-related deaths among women worldwide. Early diagnosis is an important approach leading to good prognosis and a high survival rate. However, the morbidity and mortality from breast cancer are still high. Therefore, the investigation of new prognostic factors is necessary for breast cancer patients [1,2].

Single nucleotide polymorphisms (SNPs), located in genes, which code proteins involved in the regulation of cell cycle and apoptosis, can cause dysregulation of essential cellular processes by affecting protein expression or activity resulting in uncontrolled cell growth [3]. Previous studies indicated that *TP53*, *BBC3*, *CCND1* and *EGFR* genes play important roles in the pathogenesis of breast cancer [4,5,6,7]. However, the roles of most SNPs in these genes have not been fully defined.

A tumor suppressor protein p53 (encoded by *TP53*) is involved in regulation of cell growth, apoptosis, DNA recombination, damage repair. Its response to stress, such as hypoxia, metabolite or oncogene activation, is a key factor in maintaining genomic stability. It is believed that polymorphisms in *TP53* gene may influence its functional effects. The most common polymorphism is rs1042522. This SNP leads to the transversion of cytosine to guanine resulting in the substitution from proline (Pro) to arginine (Arg) at codon 72 [4,8]. Studies have shown that proline is associated with better control of cell cycle and DNA repair, compared to the arginine, which is associated with much faster and more efficient apoptosis [9].

P53 upregulated modulator of apoptosis (PUMA) is a pro-apoptotic protein, also known as Bcl-2-binding component 3 (BBC3). BBC3 is a critical mediator of apoptosis in response to p53 tumor suppressor and other apoptotic stimuli, such as deregulated oncogene expression, toxins and the deficiency of growth factors [10]. *BBC3* rs2032809 polymorphism causes the conversion of adenine to guanine in the gene promoter. Although Zhou et al. [11] suggested that G allele significantly reduces the binding affinity of any transcriptional factor to the *BBC3* promoter and slightly reduces BBC3 expression, the functional effect of this SNP is poorly understood yet.

Cyclin D1 is a key regulator in controlling the cell cycle, promoting the cell transition from the G phase to the S phase. This protein is encoded by a highly polymorphic *CCND1* gene [12]. The rs9344 is a common SNP that is located at codon 241 and it results in the alternative splicing [13].

*EGFR* gene encodes the epidermal growth factor receptor, which promotes cell cycle progression by activating signal transduction pathways. Several SNPs are located in *EGFR*, including rs2227983. This polymorphism includes a guanine to adenine transition leading to an Arginine (Arg) to Lysine (Lys) substitution at codon 521 [14,15]. Morioi et al. [16] reported that A allele is associated with reduced function of the receptor because the substitution is located in CR2 domain and results in a lower affinity to ligands, reduced growth stimulation and induction of proto-oncogenes *MYC*, *FOS* and *JUN.*

Therefore, the aim of this study was to analyze *TP53* rs1042522, *BBC3* rs2032809, *CCND1* rs9344, *EGFR* rs2227983 polymorphisms and their associations with tumor clinicopathological features and clinical outcomes in breast cancer patients.

## 2. Materials and Methods

### 2.1. Study Subject

A total of 171 Lithuanian women diagnosed with breast cancer were enrolled in this study. Blood samples were collected at the Hospital of Lithuanian University of Health Sciences Kaunas Clinics between 2014 and 2018. The study group consisted of females aged between 30 and 75 years (mean ± SD: 47.49 ± 10.14). Clinicopathological data was collected from medical records with the help of oncologists. The patients’ exclusion criteria were as follows: Other malignancies and significant comorbidities, poor performance status and incomplete medical documentation. The age at the time of diagnosis, differentiation degree (G), tumor size (T), estrogen (ER) and progesterone (PR) receptors status, human epidermal growth factor receptor 2 (HER2) status, lymph node involvement (N), presence of disease progression, development of metastasis and patients’ death were considered as clinicopathological features in this analysis.

The study was approved by Kaunas Regional Biomedical Research Ethical Committee (protocols No. BE-2-10 and No. P1-BE-2-10/2014). A written informed consent was obtained from all the participants.

### 2.2. DNA Extraction and Genotyping

Genomic DNA was extracted from peripheral blood leukocytes using DNA extraction kit (Thermo Fisher Scientific Baltics, Vilnius, Lithuania) following the manufacturer‘s instructions. DNA was stored at −20 °C until PCR.

Based on the studies of other authors, modified protocols were used for genotyping [11,17,18,19]. For all polymorphisms, the PCR reaction was carried out at a final volume of 25 µL containing distilled water (dH2O), 1× DreamTaq Buffer, 0.24 pmol/µL of each primers, 0.2 mM of each dNTPs, 0.02 U of DreamTaq DNA polymerase (Thermo Fisher Scientific Baltics, Vilnius, Lithuania) and template DNA. The negative control was added in order to check for contamination of components in each experiment. The primers, PCR thermal conditions and products size are summarized in Table 1. The amplified PCR products were analyzed by electrophoresis in 2% agarose gel and were visualized by staining with 0.5 µg/mL ethidium bromide under UV light.

Polymerase chain reaction-restriction fragment length polymorphism (PCR-RFLP) assay was used to genotype polymorphisms in *BBC3*, *TP53*, *EGFR* and *CCND1* genes. Following PCR the products were digested with *BstU*I (for rs1042522), *MboI*I (rs2032809), *Msp*I (for rs9344) and *Mva*I (for rs2227983) restriction enzymes, respectively. The digestion reactions were incubated at 37 °C for 2–16 h. After the digestion, *TP53* rs1042522 C allele was indicated as a 199 bp fragment, while G allele resulted in 86 and 113 bp fragments. For the *BBC3* rs2032809 A allele, the enzyme cut the PCR product into 34 and 157 bp fragments. The G allele remained uncut. For the *CCND1* rs9344 G allele, the PCR product was cut into 145 and 22 bp fragments, A allele was not cleaved by the enzyme. The G allele of *EGFR* rs2227983 resulted in 67, 50 and 38 bp fragments, while A allele was cut into 117 and 38 bp fragments. The products of the digestion reaction were separated by 3% agarose gel electrophoresis and visualized under UV light after ethidium bromide staining.

### 2.3. Statistical Analysis

The Hardy-Weinberg equilibrium (HWE) was analyzed for differences in genotypes distribution using a Chi-square test. Pearson‘s Chi-square test was used to estimate the association between genotypes and clinicopathological features. Monte Carlo *p* value was assessed when >25% of cells had expected count less than 5. For all significant associations, univariate and multivariate regression analysis was used to calculate the odds ratios (ORs) with 95% confidence intervals (95% CI). A Bonferroni correction was applied in association analysis for multiple comparison. *p* values < 0.05 were considered statistically significant, after Bonferroni corrections—*p* < 0.013.

The clinical outcomes, including overall survival (OS), progression-free survival (PFS) and metastasis-free survival (MFS), were also analyzed in the study. The OS was measured from the date of diagnosis until the date of death or last follow-up. PFS and MFS were calculated from the date of diagnosis till the event—local and systematic disease spread or distant metastasis, respectively, or the most recent follow-up. Survival curves were generated using the Kaplan-Meier method based on a log-rank test. Univariate and multivariate Cox proportional hazard models were used to perform the hazard ratios (HRs). *p* values < 0.05 were considered statistically significant. Three models were used for multivariate analysis: Model no. 1 (adjusted for age at the time of diagnosis), Model no. 2 (adjusted for age at the time of diagnosis, differentiation degree, T, N) and Model no. 3 (adjusted for age at the time of diagnosis, differentiation degree, T, N and ER, PR, HER2 status).

The Statistical Package for the Social Sciences (SPSS) version 20.0 statistical software (SPSS Inc., Chicago, IL, USA) was used to perform all statistical analysis.

## 3. Results

### 3.1. Subjects Characteristics

Clinicopathological features of this study population are shown in Table 2. Briefly, the majority of patients were 50 years old or younger, with a mean age of diagnosis of 42.7 ± 5.6 years. A significant or moderately (grade G1 or G2, respectively) differentiated tumor was found in most cases (77.2%). Tumor size ranged from 0 to 5 cm and the majority had smaller tumor (≤2 cm) (66.7%). Moreover, our results showed that 67.8% and 59.1% of patients were positive for ER and PR, respectively, while HER2 expression was found only in 18.7% of cases. The lymph node involvement was identified for 38.6% of patients.

During a follow-up period, disease progression was confirmed for 32 (18.7%) patients, while metastasis was identified for 27 (15.8%) patients. For the studied population, the median PFS and MFS were 38 and 41 months, respectively. Moreover, overall 22 (12.9%) patients died after a median follow-up of 88 months (all of them developed disease progression and metastasis).

### 3.2. The Distribution of TP53 rs1042522, BBC3 rs2032809, CCND1 rs9344 and EGFR rs2227983 Genotypes in Patients with Breast Cancer

In this study, all polymorphisms were found to be in Hardy-Weinberg equilibrium. The genotype distribution of the *TP53* rs1042522 was as follows: 5.8% CC, 35.1% CG, and 59.1% GG. For the *BBC3* rs2032809, the frequencies of AA, AG and GG genotypes were 22.2%, 48.0%, 23.4%, respectively. The distribution of *CCND1* rs9344 among GG, GA and AA genotypes was 26.9%, 50.9% and 22.2%, respectively. The *EGFR* rs2227983 GG genotype was identified for 62.0%, GA for 32.7%, and AA for 5.3% of patients. 

### 3.3. The Associations between TP53 rs1042522, BBC3 rs2032809, CCND1 rs9344 and EGFR rs2227983 Polymorphisms and Clinicopathological Features

The statistical analysis was performed to determine the associations between *TP53* rs1042522, *BBC3* rs2032809, *CCND1* rs9344 and *EGFR* rs2227983 polymorphisms and clinicopathological features of breast cancer (*n* = 171). The relationship with the fact of presence of disease progression, development of metastasis and patient’s death was also assessed. In this study *TP53* rs1042522 and EGFR rs2227983 did not show any statistically significant associations with analyzed breast cancer characteristics. Meanwhile, our results showed several significant associations between *BBC3* rs2032809 and *CCND1* rs9344 polymorphisms and clinicopathological features. The results by Pearson’s Chi-square test are mentioned below. The statistically significant results by univariate logistic regression analysis are summarized in Table 3.

Our findings revealed that *BBC3* rs2032809 had a statistically significant association with age at the time of diagnosis (*p* = 0.009) (Pearson Chi-square test), even after Bonferroni correction. The univariate logistic regression analysis showed that patients with AG and GG genotypes had increased risk of BC diagnosis at older age (>50 years) (OR = 4.808, 95% CI 1.348–17.144, *p* = 0.015; OR = 6.552, 95% CI 1.758–24.415, *p* = 0.005, respectively) compared to the patients with AA genotype (Table 3). However, the significance only remained between GG genotype and older age after Bonferroni correction. In addition, G allele was found to be statistically associated with analyzed feature (*p* = 0.003) in the allelic model. G allele was more prevalent in the group of patients over 50 years old (OR = 5.421, 95% CI 1.578–18.620, *p* = 0.007) compared with G allele non-carriers (Table 3). In a multivariate logistic regression analysis, G allele was considered as covariate with other factors (T, N, G, ER, PR, HER2). The results showed that association remained significant in Model no. 2 (*p* = 0.006) and Model no. 3 (*p* = 0.004) (Table 4). In addition, association between G allele and older age also remained significant after Bonferonni correction.

Furthermore, statistically significant associations were identified between *BBC3* rs2032809 and presence of disease progression (*p* = 0.001), development of metastasis (*p* = 0.003) and patients’ death (*p* = 0.001). After Bonferroni correction, associations remained statistically significant. The holders of AG genotype had a higher risk for disease progression than those with AA (OR = 5.409, 95% CI 1.524–19.205, *p* = 0.009) genotype. Moreover, AG genotype was associated with higher risk for metastasis in comparison to AA genotype (OR = 4.246, 95% CI 1.184–15.222, *p* = 0.026). Compared with AA genotype, the patients with AG genotype had even 12 times higher probability for death (OR = 11.762, 95% CI 1.514–91.379, *p* = 0.018 (Table 3). In a multivariate logistic regression analysis the associations with disease progression, metastasis and death remained significant in Model no. 1, Model no. 2 and Model no. 3 (Table 4). In univariate logistic regression analysis a statistically significant association remained only between AG genotype and progression, while no significance was attained between AG genotype and metastasis or death after Bonferroni correction. However, in multivariate analysis, associations between AG genotype and disease progression, metastasis and death remained statistically significant even after Bonferroni correction. Since statistically significant associations were estimated between heterozygous genotype of *BBC3* rs2032809 and analyzed breast cancer characteristics, the analysis of allelic models was not performed. Evaluating the association of two alleles andthe analyzed feature is complicated.

In this study, the association between CCND1 rs9344 and BC characteristics was found only in the allelic model. It was determined that G allele was associated with tumor size (*p* = 0.037). The univariate logistic regression analysis revealed that the larger tumor size (T2) was significantly less frequently found in the carriers of G allele (OR = 0.461, 95% CI 0.220–0.964, *p* = 0.040) compared with non-carriers (Table 3). This association remained significant in all three multivariate analysis models: Following the adjustment for age at the time of diagnosis (*p* = 0.035); age at the time of diagnosis, lymph node involvement, differentiation grade (*p* = 0.041); age at the time of diagnosis, lymph node involvement, differentiation grade and tumor receptor status (*p* = 0.016) (Table 5). Although the results were statistically significant in Pearson Chi-square, and univariate and multivariate logistic regression analysis, there was no statistical significance when Bonferroni correction was applied.

### 3.4. Survival Analysis

Survival analysis was performed to assess the prognostic value of all studied polymorphisms (*n* = 171). In this study, we generated survival curves using the Kaplan-Meier method and found a few statistically significant associations (Figure 1).

*BBC3* rs2032809 was statistically associated with overall survival (OS), progression-free survival (PFS) and metastases-free survival (MFS) (log-rank, *p* = 0.000, *p* = 0.000, *p* = 0.001, respectively) (Figure 1a–c). The univariate Cox proportional hazard regression analysis showed that patients with AG genotype were more likely to have a shorter OS (HR = 14.454, 95% CI 1.934–108.040, *p* = 0.009), PFS (HR = 6.754, 95% CI 2.031–22.459, *p* = 0.002) and MFS (HR = 5.303, 95% CI 1.577–17.830, *p* = 0.007) compared to the patients with AA genotype (Table 6). In the allelic model, G allele was also significantly associated with OS (log-rank, *p* = 0.004), PFS (log-rank, *p* = 0.005) and MFS (log-rank, *p* = 0.022) (Figure 1d–f). The shorter OS (HR = 10.358, 95% CI 1.393–77.034, *p* = 0.022), PFS (HR = 4.735, 95% CI 1.438–15.593, *p* = 0.011) and MFS (HR = 3.696, 95% CI 1.110–12.303, *p* = 0.033) were determined for carriers of G allele compared with non-carriers (Table 6). The mean time of OS, PFS and MFS for *BBC3* rs2032809 polymorphism was 174 (95% CI 163–184), 157 (95% CI 143–171), 163 (95% CI 150–177) months, respectively.

The analysis revealed that *EGFR* rs2227983 was associated with PFS (log-rank, *p* = 0.038) (Figure 1g). Compared with the GG genotype of rs2227983, AA genotype was associated with shorter PFS (HR = 3.358, 95% CI 1.116–10.105, *p* = 0.031) (Table 7). In addition, G allele was found to be statistically associated with OS (log-rank, *p* = 0.030) and PFS (Log Rank, *p* = 0.011) (Figure 1h,i). Regarding Cox regression analysis, the carriers of G allele had longer OS (HR = 0.282, 95% CI 0.083–0.955, *p* = 0.042) and PFS (HR = 0.275, 95% CI 0.095–0.795, *p* = 0.017) compared with non-carriers (Table 7). For *EGFR* rs2227983, the mean time of OS was 174 months (95% CI 163–184), while in case of PFS—157 months (95% CI 143–171).

In this study, multivariate analysis was performed to identify whether genotypes or alleles of *BBC3* rs2032809 and *EGFR* rs2227983 polymorphisms are independent prognostic factors for OS, PFS and MFS in patients with breast cancer (Table 8).

Patients’ age at the time of diagnosis, T, N, G and ER, PR, HER2 receptors status were selected as covariate variables in breast cancer. For this analysis, G allele of *BBC3* rs2032809 was selected as covariate together with other factors. The association between G allele and PFS, MFS, OS remained significant in all three models (*p* < 0.05). However the PFS and MFS G alleles were related to lymph node involvement, while only in the case of OS, the G allele was independent prognostic factor. Although AA genotype of *EGFR* rs2227983 showed significant associations with PFS in the Model no. 1 (*p* = 0.031) and Model no. 2 (*p* = 0.033), no statistically significant association was found in Model no. 3 analysis (*p* = 0.060). The associations between G allele and OS, PFS also resulted in loss of significance in Model no. 3 (*p* = 0.056; *p* = 0.061, respectively). Therefore, these findings suggest that other factors could be more important for those associations.

## 4. Discussion

To our knowledge, this is the first study that investigated the associations between *TP53* rs1042522, *BBC3* rs2032809, *CCND1* rs9344, *EGFR* rs2227983 polymorphisms and BC clinicopathological features and prognosis in Lithuanian population. In this study, we identified several statistically significant associations between *BBC3* rs2032809, *CCND1* rs9344, *EGFR* rs2227983 and studied characteristics, while no associations were found for *TP53* rs1042522. 

Firstly, we analyzed the associations between *TP53* rs1042522 and BC clinicopathological features and prognosis. Among the various SNPs in *TP53,* rs1042522 is the most commonly studied polymorphism in cancer epidemiology [20,21]. Several studies reported that rs1042522 may play a significant role in BC development. However, studies have revealed controversial results regarding relationship between polymorphism genotypes and BC [22,23]. In our study, this polymorphism did not show any significant associations with studied BC characteristics and patient survival. Our results are in agreement with the reports of Ayoubi et al. [21], Al-Eitanet al. [24] and Icen-Taskinet al. [25]. However, in contrast with our results, several studies showed a significant association between rs1042522 polymorphism and patient survival [26,27]. Tommiska et al. [26] and Rodrigues et al. [27] showed that patients with CC genotype had significantly poorer overall survival in Finnish, and Spanish populations, respectively.

In the present study, we also investigated the association of *BBC3* rs2032809 and BC features. To our knowledge, only few studies have investigated the association of rs2032809 and cancer in general [11,28]. Therefore, the role of this polymorphism is still not fully understood. Schuetz et al. [28] investigated the associations with status of ER, PR and HER2. However, their results did not reach statistical significance. In contrast, our findings revealed a statistically significant association between rs2032809 and age at the time of diagnosis, presence of progression, development of metastasis and mortality. The results revealed that rs2032809 AG and GG genotypes (also G allele) were statistically significantly associated with older age at diagnosis. Moreover, the heterozygous rs2032809 genotype was associated with increased risk of progression, metastasis and mortality compared with AA genotype. In the survival analysis, the AG genotype was associated with shorter OS, PFS and MFS. The analysis of allelic models showed that shorter PFS, MFS and OS were significantly associated with presence of G allele. It is likely that AG genotype and G allele may lead to worse prognosis for breast cancer patients. The studies show that very low levels of BBC3 expression are maintained in the cells normally. However, the response to apoptotic signals increases BBC3 level and induces BBC3 as a potential tumor suppressor. Therefore, it is suggested that the activity of BBC3 expression may be transcriptionally regulated [29]. In the study by Zhou et al. [11], G allele was significantly associated with lower binding affinity of any transcriptional factors to the gene promoter. Whereas, the A allele was found to have significantly stronger binding to the nuclear proteins. Based on this knowledge and our study results, we suggest that *BBC3* rs2032809 AG genotype in the gene promoter may potentially affect the regulation of BBC3 activity, leading to the dysregulation of apoptosis. Nevertheless, the influence of genotypes and alleles remain unclear and more research is needed to understand the influence of *BBC3* rs2032809 polymorphism.

Several studies observed that *CCND1* rs9344 is associated with breast cancer risk, but the relationship with breast cancer characteristics is still unknown [30]. In this study, we found that rs9344 G allele is related with tumor size. The results showed that the T2 BC was less common in the carriers of G allele. However, Bonferroni correction showed that an association is non-significant. To date, there are no data on the association of G allele with BC clinicopathological features and prognosis, but several studies revealed associations exist with the A allele [31,32]. It is known that an optimal splice donor site contains G allele and produce complete transcript (transcript-a), while the A allele results are in the incomplete transcript (transcript-b). The presence of the A allele results in alternative splicing, which modify an action of cyclin D1 in the cells by increasing the level of transcript-b that encodes a CCND1b protein with an altered C-terminal domain. On the one hand, the phosphorylation ability of retinoblastoma, which interacts with transcript-a and is necessary for the G1-S transition, is reduced. On the other hand, due to longer half-life and G1-S checkpoint bypass, the transcript-b results in an overexpression of CCND1b [32]. Some studies have reported that increased CCND1b expression is associated with poorer prognosis in various cancers [13,33]. Absenger et al. [33] indicated that a shorter MFS was associated with A allele in patients with colon cancer. In the study by Qiu et al. [13], the CCND1 overexpression was significantly associated with poor OS, lymph node involvement and distant metastasis in patients with colorectal cancer. These data and our results suggest that G allele seem to relate to better breast cancer prognosis.

In the present study, we did not find significant associations between *EGFR* rs2227983 and clinicopathological features. In contrast, some studies showed that rs2227983 was statistically associated with differentiation degree (tumour grade) and lymph node involvement, which was not found in our study [19,34,35]. Interestingly, Sobral-Leite et al. [34] reported that a poorly differentiated tumor was more common in patients with GA and AA genotypes. While, in the study by Kallel et al. [19], a poorly differentiated tumor was more frequently diagnosed in patients who had GG genotype. Furthermore, Kallel et al. [19] and Leite et al. [35] demonstrated GA and GG genotypes to be associated with lymph node involvement. In the study by Hsieh et al. [36], A allele was found to be associated with better tumor differentiation and less common lymph node involvement. Furthermore, in the survival analysis, we determined associations between AG genotype and shorter PFS in univariate logistic regression analysis. The carriers of G allele were less likely to have longer PFS and OS. Unfortunately, our findings resulted in a loss of significance in a multivariate analysis. Therefore, we hypothesize that other BC prognostic factors could be more important for the abovementioned associations. The similar results were obtained by Zhang et al. [37]. However, in contrast to our study, their results remained significant in the multivariate analysis. Therefore, despite the controversial results from various studies, A allele seems to be associated with better clinicopathological features and prognosis. 

There are several limitations in this study. First, our results may have been affected by the limited sample size. Secondly, there is a lack of information about the functions and underlying mechanisms on polymorphisms, especially rs2032809, in breast cancer pathogenesis. Using Bonferroni correction, some associations resulted in loss of significance, but this cannot be ignored. The main issue of this correction is that the interpretation of results depends on the number of other test performed. In some cases, really important differences may be insignificant because of the increased likelihood of type II errors.

## 5. Conclusions

In conclusion, our results showed that *BBC3* rs2032809 was associated with breast cancer phenotype and survival. Therefore, it could be applied as potential marker for breast cancer prognosis. Nevertheless, more detailed studies on a larger cohort are recommended to confirm our findings.

## Figures and Tables

**Figure 1 diagnostics-11-01419-f001:**
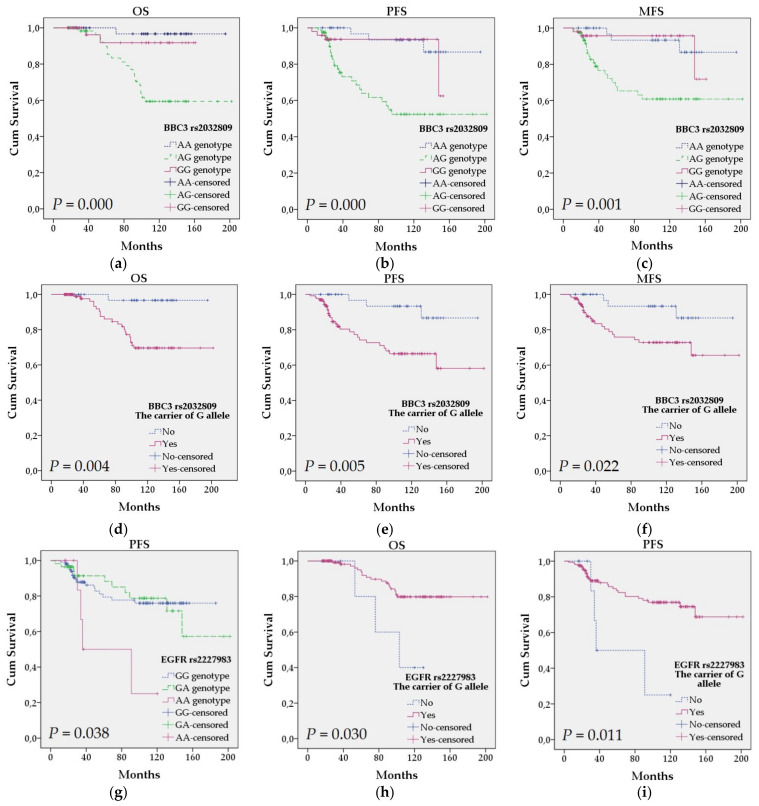
Kaplan-Meier survival curves for OS, PFS and MFS in patients with breast cancer according to the *BBC3* (rs2032809) and *EGFR* (rs2227983) polymorphisms (*n* = 171). (**a**–**c**) Patients with AG genotype of rs2032809 were at increased risk for shorter OS, PFS and MFS (*p* = 0.000, *p* = 0.000, *p* = 0.001, respectively). (**d**–**f**) A significantly shorter OS, PFS and MFS (*p* = 0.004, *p* = 0.005, *p* = 0.022, respectively) were estimated for the carriers of G allele of rs2032809. (**g**) Patients with AA genotype of rs2227983 were at increased risk for shorter PFS (*p* = 0.038). (**h**,**i**) The carriers of G allele were less likely to have a risk for OS (*p* = 0.03) and PFS (*p* = 0.011). *p* values were obtained by the log-rank test. OS = Overall survival, PFS = Progression-free-survival, MFS = Metastases-free-survival.

**Table 1 diagnostics-11-01419-t001:** Primer sequences, PCR thermal conditions and products size.

Gene, SNP	Primers Sequences	Annealing Temperature	Cycles of PCR	Size of PCR Product
***TP53* rs1042522** ^1^				
forward primer:	5′-TTGCCGTCCCAAGCAATGGATGA-3′	61.8 °C	40	199 bp
reverse primer:	5′-TCTGGGAAGGGACAGAAGATGAC-3′			
***BBC3* rs2032809** ^2^				
forward primer:	5′-GAATAATCGGGGAAAGCGAAAGAAG-3′	58 °C	35	191 bp
reverse primer:	5′-AGTGTGGGGCTGGCTGAGTAAG-3			
***CCND1* rs9344** ^3^				
forward primer:	5′-GTGAAGTTCATTTCCAATCCGC-3′	53 °C	40	167 bp
reverse primer:	5′-GGGACATCACCCTCACTTAC-3			
***EGFR* rs2227983** ^4^				
forward primer:	5′-TGCTGTGACCCACTCTGTCT-3′	63 °C	40	155 bp
reverse primer:	5′-CCAGAAGGTTGCACTTGTCC-3			

Primer sequences have been described by Jin et al. [17] ^1^, Zhou et al. [11] ^2^, Liu et al. [18] ^3^, Kallel et al. [19] ^4^.

**Table 2 diagnostics-11-01419-t002:** Clinicopathological features of the patients with breast cancer (*n* = 171).

Clinicopathological Features	*n*	%
**Age (range 30–75)**		
≤50 years	128	74.9
>50 years	43	25.1
**Differentiation degree (G)**		
G1 (well differentiated)	12	7
G2 (moderately differentiated)	120	70.2
G3 (poorly differentiated)	39	22.8
**Tumor size (T)**		
T1 (≤2 cm)	114	66.7
T2 (2–5 cm)	57	33.3
**Estrogen receptor (ER)**		
Negative	55	32.2
Positive	116	67.8
**Progesterone receptor (PR)**		
Negative	70	40.9
Positive	101	59.1
**Human epidermal growth factor receptor 2 (HER2)**		
Negative	139	81.3
Positive	32	18.7
**Lymph node (N)**		
N0 (negative)	105	61.4
N1 (positive)	66	38.6
**The presence of disease progression**		
Absent	32	18.7
Present	139	81.3
**Development of metastasis**		
Absent	27	15.8
Present	144	84.2
**Death**		
Absent	22	12.9
Present	149	87.1

**Table 3 diagnostics-11-01419-t003:** The statistically significant associations between genotypes or alleles and clinicopathological features (*n* = 171).

Gene, SNP	Genotype or Allele	Feature	OR	95% CI	*p*
*BBC3* rs2032809	AG versus AA (ref.) GG versus AA (ref.) The carrier of G allele versus non-carrier	Age at the time of diagnosis	4.808 6.552 5.421	1.348–17.144 1.758–24.415 1.578–18.620	0.015 * 0.005 0.007
	AG versus AA (ref.)	Disease progression	5.409	1.524–19.205	0.009
	AG versus AA (ref.)	Metastasis	4.246	1.184–15.222	0.026 *
	AG versus AA (ref.)	Death	11.762	1.514–91.379	0.018 *
*CCND1* rs9344	The carrier of G allele versus non-carrier	Tumor size	0.461	0.220–0.964	0.040 *

* Statistically significant *p* values that lost significance after Bonferroni correction.

**Table 4 diagnostics-11-01419-t004:** Multivariate logistic regression analysis. The adjusted odds ratio for association between BBC3 rs2032809 genotypes or alleles and clinicopathological features (*n* = 171).

Gene, SNP	Dependent	Covariates	Model No. 1			Model No. 2			Model No. 3		
			Odds	95% CI	*p*	Odds	95% CI	*p*	Odds	95% CI	*p*
*BBC3* rs2032809	Older age (≥50 years)	**The carrier of G allele vs. non-carrier**	-	-	-	5.838	1.652–20.632	0.006	6.554	1.799–23.880	0.004
		Age ^1^	-	-	-	-	-	-	-	-	-
		T (T2 vs. T1)				1.273	0.521–3.115	0.596	1.388	0.547–3.520	0.490
		N (Pos vs. Neg)				0.287	0.112–0.735	0.009	0.250	0.094–0.663	0.005
		G (G3 vs. G1+G2)				0.201	0.056–0.728	0.015	0.360	0.090–1.444	0.150
		ER (Pos vs. Neg)							3.334	0.930–11.950	0.064
		PR (Pos vs. Neg)							1.252	0.410–3.827	0.693
		HER2 (Pos vs. Neg)							1.502	0.506–4.463	0.464
	Disease progression	**AG vs. AA (ref.)**	7.892	2.178–28.593	0.002	8.165	2.219–30.048	0.002	7.415	1.961–28.045	0.003
		Age ^1^	0.056	0.007–0.432	0.006	0.064	0.008–0.507	0.009	0.068	0.008–0.552	0.012
		T (T2 vs. T1)				0.754	0.289–1.966	0.564	0.752	0.285–1.984	0.564
		N (Pos vs. Neg)				2.333	0.923–5.895	0.073	2.327	0.905–5.985	0.080
		G (G3 vs. G1+G2)				0.846	0.315–2.274	0.740	0.655	0.206–2.085	0.474
		ER (Pos vs. Neg)							1.195	0.385–3.708	0.758
		PR (Pos vs. Neg)							0.573	0.174–1.893	0.361
		HER2 (Pos vs. Neg)							0.740	0.215–2.540	0.632
	Metastasis	**AG vs. AA (ref.)**	5.917	1.622–21.593	0.007	5.952	1.606–22.050	0.008	5.601	1.446–21.694	0.013
		Age ^1^	0.075	0.010–0.580	0.013	0.090	0.011–0.723	0.023	0.094	0.011–0.775	0.028
		T (T2 vs. T1)				1.128	0.427–2.981	0.808	1.105	0.411–2.970	0.843
		N (Pos vs. Neg)				2.373	0.900–6.259	0.081	2.312	0.856–6.246	0.098
		G (G3 vs. G1+G2)				0.900	0.324–2.496	0.839	0.687	0.208–2.272	0.538
		ER (Pos vs. Neg)							1.271	0.399–4.043	0.685
		PR (Pos vs. Neg)							0.509	0.149–1.738	0.281
		HER2 (Pos vs. Neg)							0.920	0.263–3.224	0.896
	Death	**AG vs. AA (ref.)**	17.100	2.178–134.257	0.007	17.106	2.158–135.56	0.007	19.723	2.257–172.322	0.007
		Age ^1^	0.000	0.000	0.997	0.000	0.000	0.997	0.000	0.000	0.997
		T (T2 vs. T1)				1.112	0.380–3.254	0.847	1.068	0.352–3.243	0.907
		N (Pos vs. Neg)				2.141	0.731–6.270	0.165	1.922	0.629–5.869	0.251
		G (G3 vs. G1+G2)				1.316	0.447–3.868	0.618	1.022	0.287–3.643	0.973
		ER (Pos vs. Neg)							1.604	0.446–5.765	0.469
		PR (Pos vs. Neg)							0.379	0.095–1.513	0.169
		HER2 (Pos vs. Neg)							1.802	0.456–7.119	0.401

^1^ Age at the time of diagnosis (>50 years vs. ≤50 years), Pos vs. Neg = Positive versus Negative, OR = Odds ratio, CI = Confidence interval; Model No. 1—adjusted for age at diagnosis; Model No. 2—adjusted for age at diagnosis, differentiation degree, N;Model no. 3—adjusted for age at diagnosis, differentiation degree, N and ER, PR, HER2 status.

**Table 5 diagnostics-11-01419-t005:** Multivariate logistic regression analysis. The adjusted odds ratio for association between CCND1 rs9344, clinicopathological features and tumor size (*n* = 171).

Gene, SNP	Dependent	Covariates	Model No. 1			Model No. 2			Model No. 3		
			Odds	95% CI	*p*	Odds	95% CI	*p*	Odds	95% CI	*p*
*CCND1* rs9344	Larger T (T2)	**The carrier of G allele vs. non-carriers**	0.450	0.214–0.946	0.035 *	0.434	0.195–0.965	0.041 *	0.359	0.156–0.826	0.016 *
		Age ^1^	0.680	0.315–1.472	0.328	1.222	0.520–2.869	0.646	1.196	0.495–2.889	0.691
		N (Pos vs. Neg)				4.164	2.022–8.573	0.000	3.737	1.775–7.870	0.001
		G (G3 vs. G1+G2)				2.194	0.982–4.906	0.056	1.873	0.725–4.842	0.195
		ER (Pos vs. Neg)							1.982	0.711–5.525	0.191
		PR (Pos vs. Neg)							0.380	0.140–1.029	0.057
		HER2 (Pos vs. Neg)							1.308	0.520–3.287	0.569

^1^ Age at the time of diagnosis (>50 years vs. ≤50 years), Pos vs. Neg = Positive versus Negative, OR = Odds ratio, CI = Confidence interval; Model No. 1—adjusted for age at diagnosis; Model No. 2—adjusted for age at diagnosis, differentiation degree, N; Model no. 3—adjusted for age at diagnosis, differentiation degree, N and ER, PR, HER2 status; * Statistically significant *p* values that lost significance after Bonferroni correction.

**Table 6 diagnostics-11-01419-t006:** Cox’s univariate models for OS, PFS and MFS adjusted for *BBC3* rs2032809 (*n* = 171).

Gene, SNP	Genotype or Allele	Features	HR	95% CI	*p*
*BBC3* rs2032809	AG versus AA (ref.) The carrier of G allele versus non-carrier	OS	14.454 10.358	1.934–108.040 1.393–77.034	0.009 0.022
	AG versus AA (ref.) The carrier of G allele versus non-carrier	PFS	6.754 4.735	2.031–22.459 1.438–15.593	0.002 0.011
	AG versus AA (ref.) The carrier of G allele versus non-carrier	MFS	5.303 3.696	1.577–17.830 1.110–12.303	0.007 0.033

HR = Hazard ratio, CI = Confidence interval.

**Table 7 diagnostics-11-01419-t007:** Cox’s univariate models for OS, PFS and MFS adjusted for *EGFR* rs2227983 (*n* = 171).

Gene, SNP	Genotype or Allele	Features	HR	95% CI	*p*
*EGFR* rs2227983	The carrier of G allele versus non-carrier	OS	0.282	0.083–0.955	0.042
	AA versus GG (ref.) The carrier of G allele versus non-carrier	PFS	3.358 0.275	1.116–10.105 0.095–0.795	0.031 0.017

HR = Hazard ratio, CI = Confidence interval.

**Table 8 diagnostics-11-01419-t008:** Cox’s multivariate models: OS, PFS and MFS adjusted for BBC3 rs2032809, OS and PFS adjusted for EGFR rs2227983 (*n* = 171).

Gene, SNP	Dependent	Covariates	Model No. 1			Model No. 2			Model No. 3		
			Odds	95% CI	*p*	Odds	95% CI	*p*	Odds	95% CI	*p*
*BBC3* rs2032809	OS	**The carrier of G allele vs. non-carriers**	10.423	1.402–77.511	0.022	10.658	1.432–79.294	0.021	11.030	1.446–84.120	0.021
		Age ^1^	0.000	0.000	0.981	0.000	0.000	0.981	0.000	0.000	0.981
		T (T2 vs. T1)				1.318	0.553–3.140	0.533	1.214	0.483–3.050	0.680
		N (Pos vs. Neg)				2.103	0.855–5.177	0.106	1.940	0.754–4.992	0.169
		G (G3 vs. G1+G2)				1.082	0.448–5.614	0.861	0.881	0.317–2.450	0.808
		ER (Pos vs. Neg)							1.485	0.550–4.004	0.435
		PR (Pos vs. Neg)							0.447	0.151–1.323	0.146
		HER2 (Pos vs. Neg)							1.166	0.395–3.437	0.781
	PFS	**The carrier of G allele vs. non-carriers**	5.060	1.537–16.653	0.008	5.384	1.631–17.769	0.006	5.108	1.515–17.218	0.009
		Age ^1^	0.234	0.031–1.778	0.160	0.265	0.034–2.040	0.202	0.279	0.036–2.181	0.224
		T (T2 vs. T1)				0.887	0.417–1.886	0.755	0.889	0.411–1.924	0.765
		N (Pos vs. Neg)				2.340	1.098–4.985	0.028	2.363	1.089–5.126	0.030
		G (G3 vs. G1+G2)				0.860	0.393–1.882	0.705	0.756	0.301–1.895	0.550
		ER (Pos vs. Neg)							1.319	0.560–3.107	0.527
		PR (Pos vs. Neg)							0.633	0.265–1.511	0.303
		HER2 (Pos vs. Neg)							0.741	0.273–2.015	0.558
	MFS	**The carrier of G vs. non-carriers**	3.924	1.179–13.067	0.026	4.165	1.248–13.898	0.020	4.119	1.193–14.219	0.025
		Age ^1^	0.296	0.038–2.292	0.244	0.347	0.044–2.721	0.314	0.348	0.043–2.791	0.320
		T (T2 vs. T1)				1.257	0.565–2.798	0.575	1.230	0.538–2.814	0.623
		N (Pos vs. Neg)				2.383	1.033–5.496	0.042	2.328	0.985–5.504	0.054
		G (G3 vs. G1+G2)				0.862	0.371–2.000	0.729	0.752	0.277–2.039	0.576
		ER (Pos vs. Neg)							1.425	0.564–3.597	0.454
		PR (Pos vs. Neg)							0.565	0.221–1.446	0.234
		HER2 (Pos vs. Neg)							0.854	0.301–2.424	0.767
*EGFR* rs2227983	OS	**The carrier of G allele vs. non-carriers**	0.285	0.084–0.963	0.043	0.259	0.072–0.938	0.040	0.270	0.066–1.109	0.069
		Age ^1^	0.000	0.000	0.984	0.000	0.000	0.983	0.000	0.000	0.983
		T (T2 vs. T1)				0.979	0.395–2.425	0.963	0.904	0.342–2.386	0.838
		N (Pos vs. Neg)				1.609	0.655–3.952	0.299	1.466	0.556–3.864	0.439
		G (G3 vs. G1+G2)				1.437	0.555–3.720	0.455	1.206	0.388–3.749	0.746
		ER (Pos vs. Neg)							1.584	0.518–4.844	0.420
		PR (Pos vs. Neg)							0.414	0.136–1.263	0.121
		HER2 (Pos vs. Neg)							1.059	0.360–3.119	0.917
	PFS	**AA vs. GG (ref.)**	3.358	1.116–10.105	0.031	3.437	1.082–10.916	0.036	3.269	0.926–11.531	0.066
		Age ^1^	0.338	0.044–2.600	0.298	0.405	0.052–3.169	0.389	0.413	0.052–3.258	0.401
		T (T2 vs. T1)				0.877	0.392–1.958	0.748	0.898	0.394–2.046	0.798
		N (Pos vs. Neg)				2.036	0.968–4.281	0.061	1.997	0.914–4.364	0.083
		G (G3 vs. G1+G2)				1.011	0.442–2.309	0.980	0.838	0.317–2.215	0.722
		ER (Pos vs. Neg)							1.354	0.522–3.514	0.533
		PR (Pos vs. Neg)							0.584	0.227–1.503	0.265
		HER2 (Pos vs. Neg)							0.617	0.229–1.667	0.341
		**The carrier of G allele vs. non-carriers**	0.302	0.104–0.877	0.028	0.297	0.098–0.897	0.031	0.327	0.098–1.094	0.070
		Age ^1^	0.339	0.044–2.603	0.298	0.405	0.052–3.169	0.389	0.415	0.053–3.269	0.403
		T (T2 vs. T1)				0.868	0.395–1.911	0.726	0.872	0.389–1.955	0.739
		N (Pos vs. Neg)				2.039	0.970–4.284	0.060	2.018	0.927–4.391	0.077
		G (G3 vs. G1+G2)				1.006	0.442–2.289	0.989	0.831	0.315–2.191	0.708
		ER (Pos vs. Neg)							1.311	0.511–3.364	0.573
		PR (Pos vs. Neg)							0.614	0.248–1.522	0.293
		HER2 (Pos vs. Neg)							0.614	0.228–1.663	0.339

^1^ Age at the time of diagnosis (>50 years vs. ≤50 years), Pos vs. Neg = Positive versus Negative, OR = Odds ratio, CI = Confidence interval; Model No. 1—adjusted for age at diagnosis; Model No. 2—adjusted for age at diagnosis, differentiation degree, N; Model no. 3—adjusted for age at diagnosis, differentiation degree, N and ER, PR, HER2 status.

## Data Availability

The data presented in this study are available from the corresponding author on reasonable request.

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
