# Peer review of "The Investigation of Associations between TP53 rs1042522, BBC3 rs2032809, CCND1 rs9344, EGFR rs2227983 Polymorphisms and Breast Cancer Phenotype and Prognosis"

_diagnostics, 2021, doi:10.3390/diagnostics11081419_

Round 1

Reviewer 1 Report

1) Please add confidence intervals for the survival rates

2) Many tests are performed, corrections for multiple testing should be applied.

3) Tables 4 and 5 should be revised, they are unreadable.

4) Add the number of available patients for each univariable and multivariable analysis.

5) Figure 1 is of poor quality. Moreover, in figures 1b-c and 1h-1i, the number of events in some groups seems to small to perform a pertinent test. Discuss this.

6) Improve the header in table 8. 

Author Response

Dear Reviewer, 

We sincerely appreciate the time and efforts contributed in revising this manuscript and providing insightful comments that were useful to improve our manuscript. 

Our responses: 

Comment: Please add confidence intervals for the survival rates.

Response: Thank you for this suggestion. We have incorporated sentences as follow:

  • “The mean time of OS, PFS and MFS for BBC3 rs2032809 polymorphism was 174 (95% CI 163-184), 157 (95% CI 143-171), 163 (95% CI 150-177) months, respectively.” (p. 9, lines 267-269)
  • “For EGFR rs2227983, the mean time of OS was 174 months (95% CI 163-184), while in case of PFS – 157 months (95% CI 143-171).” (p. 9, lines 278-279)

Comment: Many tests are performed, corrections for multiple testing should be applied.

Response: Thank you very much for your attention to the corrections that are recommended when many tests are applied. For multiple testing we applied a Bonferroni correction, where P values of less than 0.013 was considered statistically significant (Bonferroni significance threshold P = 0.05/4). After applying this correction, our results remained statistically significant only between BBC3 rs2032809 and age at the time of diagnosis, disease progression, metastasis and death. No significant difference of CCND1 rs9344 and tumor size was found after Bonferroni correction. As a result of these corrections, we made corrections to our manuscript and supplemented it with the following sentences:

  • “CCND1 rs9433 was associated with tumor size, however an association resulted in loss of significance after Bonferroni correction.” (p. 1, lines 24-25)
  • “In conclusion, BBC3 rs2032809 polymorphism was associated with breast cancer phenotype and prognosis. Therefore, it could be applied as potential markers for breast cancer prognosis.” (p. 1, lines 28-30)
  • “A Bonferroni correction was applied in association analysis for multiple comparison. P – values <0.05 were considered statistically significant, after Bonferroni corrections – P<0.013.” (p. 3, lines 126-128)
  • P values <0.05 were considered statistically significant.“ (p. 4, line 136)
  • “*Statistically significant P values that lost significance after Bonferroni correction.” (p. 5, line 180)
  • “Our findings revealed that BBC3 rs2032809 had a statistically significant association with age at the time of diagnosis (P = 0.009) (Pearson Chi-square test) even after Bonferroni correction.” (p. 5, lines 181-183)
  • “However, the significance remained only between GG genotype and older age after Bonferroni correction.” (p. 5, lines 186-187)
  • “In addition, association between G allele and older age also remained significant after Bonferonni correction.” (p. 12, lines 193-195)
  • “After Bonferroni correction, associations remained statistically significant.” (p. 7, lines 203-204)
  • “In univariate logistic regression analysis a statistically significant association remained only between AG genotype and progression, while no significance was attained between AG genotype and metastasis or death after Bonferroni correction. However, in multivariate analysis, associations between AG genotype and disease progression, metastasis and death remained statistically significant even after Bonferroni correction.” (p. 7, lines 211-216)
  • “Although the results were statistically significant in Pearson Chi-square and univariate and multivariate logistic regression analysis, there was no statistical significance when Bonferroni correction was applied.” (p. 7, lines 229-231)
  • In Tables 3 and 5, P values that lost significance after Bonferroni correction were marked with “*” (the explanation was inserted in the legend).
  • “Using Bonferroni correction, some associations resulted in loss of significance, but this cannot be ignored. The main issue of this correction is that the interpretation of results depends on the number of other test performed. In some cases, really important differences may be insignificant because of increased likehood of type II errors.” (p. 13, lines 388-392)

Comment: Tables 4 and 5 should be revised, they are unreadable.

Response: We would like to apologize for unreadable tables. To facilitate the interpretation of tables 4 and 5, we revised the tables and made some changes: the names of “Covariates” have been slightly reworded adding additional explanations in the legend; the gaps between values have been increased. Based on your observation, we also adjusted Table 8 according to Tables 4 and 5.

Comment: Add the number of available patients for each univariable and multivariable analysis

Response: Thank you for your comment. We would like to note that all patients (n = 171) were enrolled for the univariate and multivariate analysis (including analysis of associations and survival analysis), however we added the number of available patients in the text (p. 5, line 171 and p.7, line 240) and in the header of the tables 3 (p. 5, line 179), 4 (p. 6, line 197), 5 (p. 7, line 233), 6 (p. 9, line 270), 7 (p. 9, line 281), 7 (p. 9, line 287) and figure 1 (p. 8, line 251).  

Comment: Figure 1 is of poor quality. Moreover, in figures 1b-c and 1h-1i, the number of events in some groups seems too small to perform a pertinent test. Discuss this.

Response: We are grateful for your opinion. First of all, we have made some changes on the Figure 1 by improving its quality. Secondly, we agree that the number of events in some groups seems small. However, as we mentioned in the manuscript, one of limitation of this study is a limited sample size which may result in a lower number of cases. As a result, we performed the analysis based on available sample. For example in figure b, the total N of genotypes (N of events) were as follows:  AA – 40 (n = 3), AG – 82 (n = 25), GG – 49 (n = 4). Holders of AG genotipe have a relatively higher risk of an event compared with individuals having AA or GG genotype. Therefore, in our opinion the difference can be estimated.  

Comment: Improve the header in table 8.

Response: Thank you very much for this observation. We reworded the header as follows:

  • “Table 8. Cox’s multivariate models: OS, PFS and MFS adjusted for BBC3 rs2032809, OS and PFS adjusted for EGFR rs2227983.” (p. 9, lines 286-287)

Reviewer 2 Report

The manuscript entitled “The investigation of associations between TP53 rs1042522, 2 BBC3 rs2032809, CCND1 rs9344, EGFR rs2227983 polymorphisms and breast cancer phenotype and prognosis.” submitted to “Diagnostics” described the association of SNPs of P53 and related genes with breast cancer aggressiveness. This is an interesting study, designed & written well.  The data are very much useful and may add new knowledge to the field. I am in favor of the publication of this manuscript.

Author Response

Dear Reviewer, 

We sincerely appreciate the time and efforts contributed in revising this manuscript and providing insightful comments that were useful to improve our manuscript. 

Our responses: 

Comment: The manuscript entitled “The investigation of associations between TP53 rs1042522, 2 BBC3 rs2032809, CCND1 rs9344, EGFR rs2227983 polymorphisms and breast cancer phenotype and prognosis.” submitted to “Diagnostics” described the association of SNPs of P53 and related genes with breast cancer aggressiveness. This is an interesting study, designed & written well.  The data are very much useful and may add new knowledge to the field. I am in favour of the publication of this manuscript.

Response: We would like to thank you so much for your time in reviewing and evaluating our manuscript. We are grateful for the positive review.